chemical physics/computer modelling and simulation/computational chemistry

physical adsorption, chemical adsorption, defective carbon nanotube, gas molecule

**Authors for correspondence:**
Zhigang Wang
e-mail: wangzg@jlu.edu.cn
Rui-Qin Zhang
e-mail: aprqz@cityu.edu.hk

This article has been edited by the Royal Society of Chemistry, including the commissioning, peer review process and editorial aspects up to the point of acceptance.

[†]These authors contributed equally to this work.

# The nature of small molecules adsorbed on defective carbon nanotubes

Danhui Li[1,2,†], Fengting Wang[1,2,†], Zhiyuan Zhang[1,2], Wanrun Jiang[1,2], Yu Zhu[1,2], Zhigang Wang[1,2] and Rui-Qin Zhang[3,4]

[1]Institute of Atomic and Molecular Physics, and [2]Jilin Provincial Key Laboratory of Applied Atomic and Molecular Spectroscopy, Jilin University, Changchun 130012, People's Republic of China
[3]Department of Physics, Centre for Functional Photonics (CFP), City University of Hong Kong, Hong Kong SAR, People's Republic of China
[4]Beijing Computational Science Research Center, Beijing 100193, People's Republic of China

ZW, 0000-0002-3028-5196

In this work, we perform a comprehensive theoretical study on adsorption of representative 10-electron molecules $H_2O$, $CH_4$ and $NH_3$ onto defective single-walled carbon nanotubes. Results of adsorption energy and charge transfer reveal the existence of both chemical adsorption (CA) and physical adsorption (PA). While PA processes are common for all molecules, CA could be further achieved by the polar molecule $NH_3$, whose lone-pair electrons makes it easier to be bonded with the defective nanotube. Our systematic work could contribute to the understanding on intermolecular interactions and the design of future molecular detectors.

## 1. Introduction

Carbon nanotubes (CNTs), which were discovered in 1991 [1,2], have attracted enduring attention due to their unique structural, thermal, electronic and dynamic properties, which facilitate the use of CNTs in promising medical and biochemical applications. In fact, CNTs have already found applications in many fields, including hydrogen storage [3], free-radicals scavenging activity [4], chemical sensors [5], nanobiology electronics [6], functional groups adsorption substrates [7], capacitors [8] and the like.

The properties of CNTs can be affected strongly by the presence of various defects, which are usually formed during their growth process or caused by the environmental factors. The properties of defective CNTs have been systematically explored in many theoretical and experimental research projects [9–13]. By using scanning gate microscopy (SGM) and scanning impedance

microscopy (SIM), workers have studied the defects in semiconducting single-walled carbon nanotubes (SWCNTs) [12]. Theoretical calculations have shown that local structural defects, such as topological defects, vacancies, impurities and deformations can substantially modify the electronic and transport properties of SWCNTs [13]. Covalent functionalization of SWCNTs with multiplicative biological groups [14–18] based on the effects of the nanotubes' conductive properties, have also been reported [19]. For instance, oxidation-induced defects have been introduced to enhance the sensitivity of an SWCNT to chemical vapours [20]. A five-membered or otherwise configured biological group inserted into the hexagon ring in the SWCNTs has generated apparent chemical reactivity [21]. Also, in the one-dimensional topological defect consisting of octagonal and pentagonal carbon rings inserted into the SWCNT's hexagon, charge transfer [22] and field-effect doping [23] can be applied to manipulate charge carrier concentrations, enabling it to act as a conducting wire [24].

Adsorption of small molecules—such as Xe [25], $CF_4$ [26], $H_2$ [27], $CH_4$ [28], NO [29], $NH_3$ [30], $NO_2$ [31] and $O_2$ [32]—onto SWCNTs, as well as adsorption of $H_2O$ onto graphene [33] or SWCNTs, has been studied theoretically. The sensitivity of SWCNTs to these various small molecules, combined with their own inherent characteristics—such as small size, favourable electrical and mechanical properties, and high surface-area-to-volume ratios, type of defect—have led to many applications for them in various roles as functionalized materials in sensors and in many biological fields. Nevertheless, the complicated, weak interactions between these molecules and carbon materials have been widely reported, but a systematic investigation on the nature of those interactions has been lacking.

A thorough and fundamental theoretical study on small molecules adsorbing on SWCNTs, with a more advanced dispersion force correction, is needed. When a strongly oxidizing defective nanotube interacts with various molecules, it has been found to exhibit different properties from those of the perfect tube. Liu *et al.* reported that reactions replacing reactive sites arise among the 5–1DB defects with the oxynitride on the CNT [34]. Actually, it is easy to chemically bind an acetone onto a Stone–Wales defect or a vacancy of the SWCNT [35]. Such chemical adsorptions (CAs) change the electronic processes and transmission characteristics of the imperfect tube. However, research efforts into the physical adsorption (PA) which is a weak interaction of small molecules, such as $H_2O$, $CH_4$ and $NH_3$, on defective tubes are seldom reported (e.g. the adsorption energy, charge transfer, gap value, dipole moment, etc.).

Previous research studies have examined the adsorption of various molecules on graphene and revealed obviously different characteristics in Raman vibrational and ultraviolet–visible adsorption spectra of the molecules [36]. However, a systematic study on the interaction of specific molecules with different types of defective SWCNTs has yet to be conducted. In this paper, we theoretically study three commonly found representative 10-electron molecules ($H_2O$, $CH_4$ and $NH_3$) adsorbed onto four common defective SWCNT structures. Our results include the various adsorption structures, adsorption energy and charge transfer with the defective structures, compared with those values from the perfect SWCNT. The case of $NH_3$ is found to be exceptional, because it shows distinct chemical characteristics when it is adsorbed on two types of defective SWCNTs. To elucidate our results, a brief discussion has also been made on the polarity of the small gas molecules in relation to their PA or CA on SWCNTs.

# 2. Computational methods

The interactions between three representative small molecules ($H_2O$, $CH_4$ and $NH_3$) and four kinds of defective CNTs were studied with the DFTB+ code [37]. Density functional tight-binding (DFTB) is based on DFT and derived from the second-order expansion of the Kohn–Sham total energy functional as calculated within DFT under the standard tight-binding theory [38]. In order to describe the van der Waals (vdW) interaction between the small molecules and defective CNTs, we adopted the DFTB-D method, which is augmented by the empirical London dispersion energy term into the DFTB total energy [39].

The London dispersion energy is given by

$$E_{\text{dis}} = \frac{3}{2} \frac{I_\alpha I_\beta p_\alpha p_\beta}{(I_\alpha + I_\beta) R_{\alpha\beta}^6}, \tag{2.1}$$

$$C_6^{\alpha\beta} = \frac{2 C_6^\alpha C_6^\beta p_\alpha p_\beta}{p_\alpha^2 C_6^\alpha + p_\beta^2 C_6^\beta} \tag{2.2}$$

and

$$C_6^\alpha = 0.75 \sqrt{N_\alpha p_\alpha^3}, \tag{2.3}$$

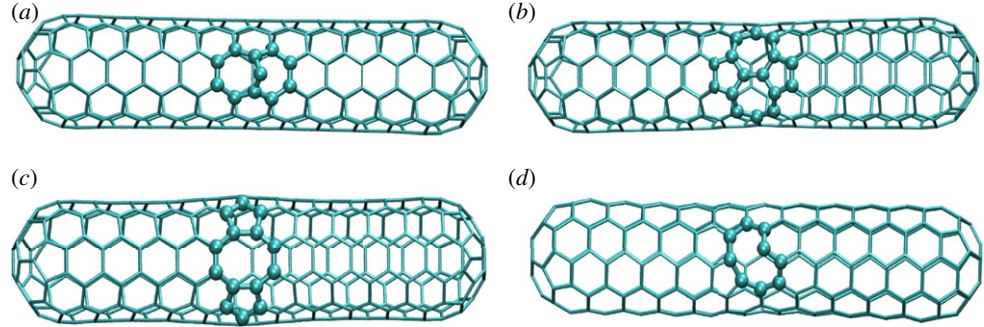

**Figure 1.** The structures of the four types of defective CNTs studied. The SWCNTs had the following defects: (*a*) an ad-atom (a), (*b*) a chiral Stone–Wale defect (b), (*c*) two missing C atoms (c) and (*d*) a monovacancy defect (d).

where $N_\alpha$ is the Slater–Kirkwood effective number of electrons. The $1/R^6$ dependence is truncated for small interatomic distances with an appropriate damping function [40]. This method has been successfully used to study organic molecules interacting with materials [41], the adsorption of water clusters on a graphene surface [42] and so forth.

To ensure the reliability of calculations, the cases of defective nanotubes with and without the cap at both ends were also considered. All of our results for the capped defective and perfect nanotubes adsorbed with $CH_4$, $H_2O$ and $NH_3$ calculated with DFTB were verified with density functional theory (DFT) calculations using PBE0-D3 functional [43,44] and 6–31 g (d, p) basis set. Conventional DFT fails in the study of weak interactions, but a newly developed method—density functional theory with empirical dispersion corrections (DFT-D)—can yield more trustworthy results. The results with DFT-D are superior because the method can compute the long-range force part, which gives more accurate data in dealing with the point charge Coulomb interaction among long-range parts; thus, it can describe PA more accurately. However, DFTB-D has an efficiency advantage in comparison with DFT-D, which is conducive to the study of more complex systems [45].

## 3. Results and discussion

In this work, we consider the interactions of $H_2O$, $CH_4$ and $NH_3$ with the pure (5, 5) SWCNT (marked with letter 'p') and four types of defective CNTs. The denotations of the four defective nanotubes are 'a' for ad-atom, 'b' for a chiral Stone–Wales (SW) defect, 'c' for the case of two missing C-atoms and 'd' for a monovacancy defect, respectively. The configurations of the CNTs with defects are presented in figure 1. There are several conformations of the same molecule adsorbed on the various CNTs. With different initial molecular-CNT conformations, we obtained 15 stable structures that are based on a large number of different initial conformations and are shown in figure 2. The results show that CA occurred when $NH_3$ adsorbed on 'a' and 'd' defective nanotubes, as shown in figure 2*a*. In figure 2*b*, the conformations for the PA case are shown. We can see several stable structures with a hydrogen atom pointing to the nanotube, which is similar with the anion-π interaction reported in the literature [46]. It should be noted that, for all the adsorbates, it is the proton instead of the heavy atom (N or O) that is pointing to the CNTs. This is because their electrons are $sp^3$ hybridized, indicating that electron lone pairs exist in the 'proton-free' directions. Such electron-negative lone pair would repulse either with the lone pairs of the dangling C or the π electrons, causing instability of the system, contrary to the electron-relative protons.

However, $NH_3$ adsorbed onto the perfect CNT reveals different properties from those of the other molecules, with the N atom staying close to the nanotube. That observation is concordant with the findings of Shirvani *et al.* [47]. In these structures, we readily found that one C atom is not coplanar with other atoms on the same six-membered rings in the 'a' and 'd' types of defective tubes, unlike the case with the 'b' and 'c' types. The structure of $CH_4$ differs in that one H atom pointing to the CNT appears in the 'a' and 'd' tubes and three H stable structures, and when $H_2O$ adsorbs on 'a' defective nanotube, the distance is the smallest. Atoms pointing to the CNT appear in the 'b' and 'c' tubes, as is the case with the 'p' tube. To summarize, we obtain different stable structures when the same molecule is adsorbed onto the various CNTs. Comparing the stable structures of the PA cases, we concluded that the distance of the $CH_4$ to 'c' defective nanotubes is the largest in all of the angles which measure the deviations of the small molecules from the *x*-axis. When $H_2O$ and $NH_3$ adsorb on 'b' defective nanotubes, the angles are larger; however, when $CH_4$ and $H_2O$ adsorb on 'd' defective

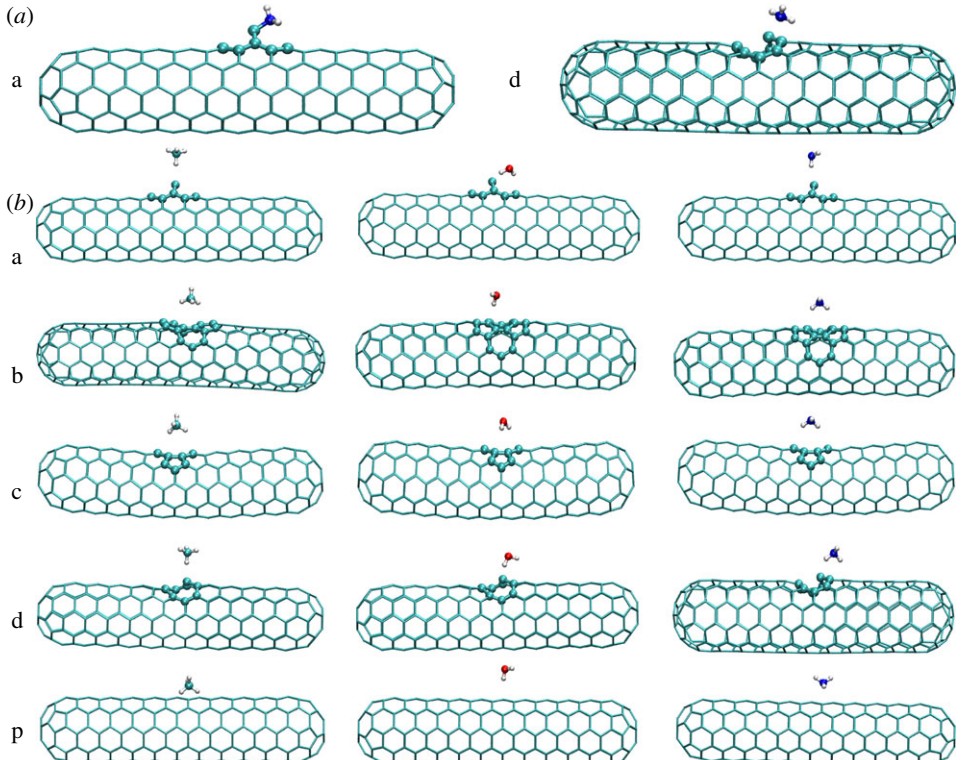

**Figure 2.** (*a*) The optimized configurations of NH$_3$ adsorbed on the 'a' and 'd' defective nanotubes for the chemical case. (*b*) The optimized configurations of the small molecules (CH$_4$, NH$_3$, H$_2$O) adsorbed on the 'a' 'b' 'c' 'd' defective and perfect nanotubes for the physical case.

and perfect nanotubes, the angles are the smallest. The data of all stable structures are provided in the electronic supplementary material.

The results of our calculations indicate that the NH$_3$ molecule behaves differently from the others. The NH$_3$ molecule exhibits both PA and CA adsorption on the 'a' and 'd' type CNTs for the different structures of the 'a' and 'd' defective nanotubes from other types, with the lone-pair electrons of the top defective C-atoms in 'a' and 'd' being more easily bonded with small molecules. By contrast, in the other defective 'b' and 'c' defective nanotubes, three electrons in the outer shell of the defective C-atoms bond with other C-atoms and the last electron forms π bonds. As for the polar molecule NH$_3$, the N-atom in the NH$_3$ takes the inequivalent sp$^3$ hybridization forming three σ bonds and one lone-pair electrons, making it is easier to bond with 'a' and 'd' defective nanotubes to form the stable structure as shown in figure 2*a*. However, the H$_2$O and CH$_4$ molecules show only PA interactions. Among them, CH$_4$ is non-polar molecule with the C-atom taking sp$^3$ equivalent hybridization and involving no lone-pair electrons while there are two lone-pair electrons in H$_2$O, making the structures of 'a' and 'd' defective nanotubes less stable.

Figure 3 reveals the direction of the charge transfer clearly for the CA case in 'a' and 'd' defective nanotubes. We can see in figure 3*a* the electron density difference of 'a' and 'd' defective nanotubes adsorbed with NH$_3$.

The isovalues of the isosurfaces are 0.004 and −0.004, respectively, for the pink and green area; the pink areas on the defective CNTs represent the electron increase but the green area expresses the charge decrease. The figures show that near the 'a' and 'd' defective CNTs the pink area is larger, revealing the direction of charge transfer to be from the defective CNTs to NH$_3$. That result is consistent with the electron density difference integral as shown in figure 3*b*, which is obtained by integrating the electron density in the $y$ and $z$ direction onto the $x$-direction, following $I(x) = \int_{-\infty}^{+\infty} \rho(x,y,z)\mathrm{d}y\mathrm{d}z$. In figure 3*b*, the vertical axis represents the integrated electron density difference and the horizontal axis is the coordinate x, ranging from 3 to 13 Å. The N atom is at $x = 9.06$ Å for 'a' defective nanotube and 9.27 Å for 'd' as shown in the electronic supplementary material, figure S1. The vertical coordinates of the points above zero represent the increment of the electron density and below zero the decrement of the electron density. We obtain the same conclusion that the charge transfer is from 'a' and 'd' defective nanotubes to NH$_3$.

In table 1, the CA values are larger than the PA values for both the adsorption energy and the charge transfer of NH$_3$ adsorbed on the 'a' and 'd' type CNTs. More specifically, the charge transfer for the

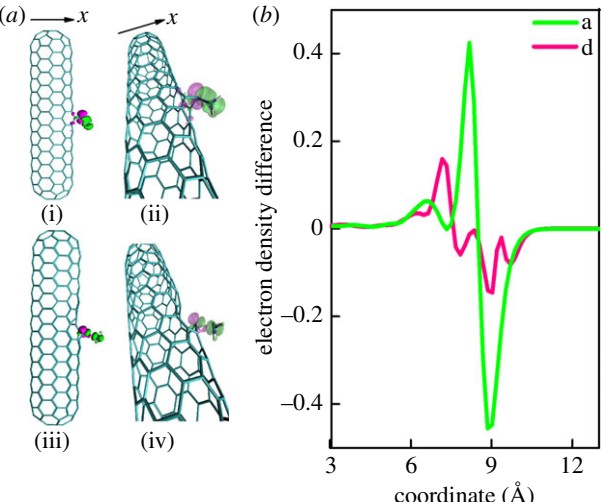

**Figure 3.** (*a*) The electron density difference of the 'a' and 'd' defective nanotubes adsorbed with $NH_3$ molecule for the CA case. (i and ii) The front and side figures of the electron density difference when $NH_3$ adsorbed on 'a' defective nanotube. (iii and iv) The front and side figures of the electron density difference when $NH_3$ is adsorbed on 'd' defective nanotube. (*b*) The electron density difference integral for the chemical adsorption that 'a' and 'd' defective nanotubes adsorbed with $NH_3$.

**Table 1.** The binding energies $\Delta E$, the charge transfer $Q$ of $NH_3$ adsorbed on 'a' and 'd' type defect CNTs for the PA and the CA, respectively.

| tube | PA | | CA | |
|------|-----|------|-----|------|
| | $\Delta E$ (meV) | $Q$ (e) | $\Delta E$ (meV) | $Q$ (e) |
| a | −78 | −0.01 | −1260 | 0.66 |
| d | −68 | 0 | −461 | 0.27 |

CA case is of the order of $10^{-1}$ e, showing a significant contrast with $10^{-3}$ e for the PA. Both of the adsorption energies for the CA cases of 'a' and 'd' defects are of the order of $10^3$ meV and $10^2$ meV, whereas the adsorption energy of both the 'a' and 'd' defects is of the order of $10^1$ meV for the PA case. However, the adsorption energy and charge transfer for adsorption on 'a' defective nanotube is larger than 'd' defective nanotube, especially for CA, for the difference of the defective C-atoms that the top defective C-atom of 'a' is most distant from the principle axis of the nanotube among the defective and perfect nanotubes, which makes $NH_3$ more easily polarized and bonded when adsorbed on 'a' defective nanotube.

Compared with the CA, the adsorption energy and the charge transfer of $H_2O$, $CH_4$ and $NH_3$ on the CNTs we calculated are much less, as shown in figure 4. The calculated PA energies vary within 400 meV. Moreover, all of them are around 100 meV except the case when $H_2O$ adsorbed on 'a' defective nanotube (321 meV). To verify the differences of this phenomenon, further electronic density and the local density of state (LDOS) analyses are carried out. It turns out that not only the electron density of the $H_2O$ adsorption in the intermolecular area is obviously larger than those of $NH_3$ and $CH_4$, but its LDOS also presents delocalization of the orbitals through the region. It could be concluded that the interaction feature of this system is not simply van de Waals intermolecular interaction (detailed information is included in the electronic supplementary material). Figure 4 also presents the charge transfers for each of the configurations. The green values represent the direction of the charge transfer that shows an electron transfer from the defective or perfect nanotubes to the small molecules (i.e. $CH_4$, $H_2O$ and $NH_3$).

Actually, the activation energy of desorption and the binding energies for small molecules adsorbed on carbon-based materials have been systematically observed in the previous experimental research [48]. The binding energy of small molecules with SWCNT is of similar magnitude to that of the PA cases above. This could also prove the reliability of our results.

Correspondingly, the pink values denote the opposite direction. It shows that the charge transfer from the perfect nanotube to $NH_3$ is of the order of $10^{-2}$ e. The same amount of charge transfer number is found from the $NH_3$ to a defective nanotube, whereas for other systems the charge transfer is almost

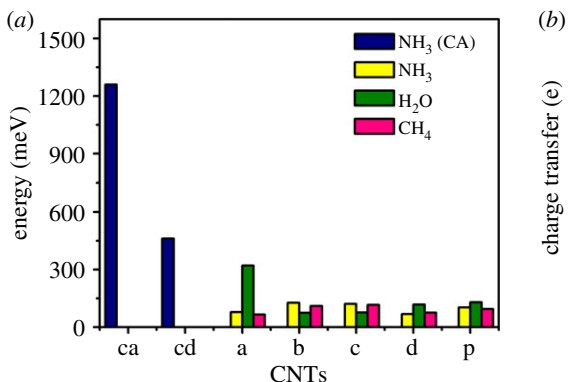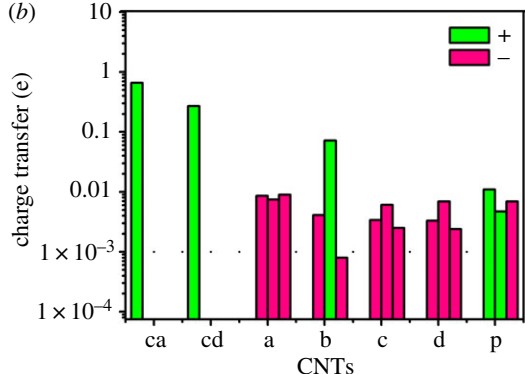

**Figure 4.** (a) The comparison of the physical and chemical adsorption energies, CA structures (blue), PA structures NH$_3$ (yellow), H$_2$O (olive), CH$_4$ (pink). (b) The comparison of the charge transfer of the PA and CA. The green (or pink) colour denotes the electron decrease (increase) or the charge increase (decrease).

negligible. The charge transfer from 'a' defective nanotube to H$_2$O is approximately $10^{-1}$ e. It is because of the lone-pair electrons of the top defective C-atom of 'a' which is the most distant from the principle axis of the nanotube among the defective and perfect nanotubes. Also, H$_2$O is polar molecular with sp$^3$ non-equivalent hybridization and the two lone-pair electrons make it easily adsorbed on the defective nanotubes especially on 'a' defective nanotube.

Furthermore, calculation of the cases for CNTs without the 'cap' were also carried out. The results show that when the 'cap' of the defective nanotubes is removed, the calculations still support the similar conclusions for the chemical case and PA cases, except for the 'c' defective nanotube whose band gap becomes larger when the cap is removed.

In order to show the electronic density-based properties of the adsorption structure, electronic density difference analyses under DFT method were carried out. As expected, electronic density differences of the CA structures are much larger than those of the PA ones, consistent with the results of charge transfer analyses above. The details of electronic density difference are provided in the electronic supplementary material.

Furthermore, we also calculated the energies of all the dissociative adsorption cases with DFTB-D. By comparing the energies of all dissociative adsorptions stable structures with the corresponding coordination complexes, we concluded that dissociative adsorptions for 'a' and 'd' defective nanotubes are possible and further calculated their adsorption energies. The comparison of their local density states is also provided, from which we can see that the distribution of the density states is more uniform in the dissociative adsorption ones than the coordination complexes, and the delocalization of the orbital is weaker. We also performed molecular dynamic simulations of the possible dissociative adsorption cases. All the simulations were performed at room temperature. However, we have not observed any dissociative adsorptions, indicating that they do not easily occur. All the figures of the structures and energy data are provided in the electronic supplementary material.

Besides, there are many studies in the literature certifying the accuracy of the DFTB method which is compared favourably with DFT [36,45,49,50]. And the 'd' defective nanotubes adsorbed chemically with NH$_3$ calculated with DFT have been reported [30], which confirms the results of our work further. The comparisons of the adsorption energies and charge transfer values of the CA cases in which NH$_3$ adsorbs on the 'a' and 'd' defective nanotubes calculated with DFTB and DFT are provided in the electronic supplementary material, tables S1 and S2. We obtained the same conclusions that there are obvious charge transfer and much larger adsorption energy when NH$_3$ adsorbs on the 'a' and 'd' defective nanotubes chemically. Because of the similarity in the results obtained with DFT and DFTB (the calculation speed of DFTB is three magnitudes faster than DFT), it can be used in the simulation and calculation of large molecules system [41,51] in the future.

## 4. Conclusion

In this work, we systematically studied the adsorption between common 10-electron small molecules and CNTs with different defects. The reliability of the results is ensured by comparing the DFTB and DFT methods. Our calculated adsorption energy, charge transfer, energy gap and dipole moment of the studied systems reveal that PA is the common process, except with NH$_3$, which undergoes both PA and

CA due to its large polarity and apparent charge transfer. CA occurs for $NH_3$ on the defective 'a' and 'd' defective nanotube involving considerable adsorption energies and charge transfer because $NH_3$ can be easily polarized. On the contrary, PAs take place for other small molecules such as $CH_4$, $H_2O$ and $NH_3$ on the 'a', 'b', 'c', 'd' defective and perfect nanotubes with negligible adsorption energies and charge transfer.

The interactions that we determine between those gas molecules and the defective nanotubes are very important for identifying a specific gas such as $NH_3$ based on the obvious different charge transfer and adsorption energies of the CA and PA. The mechanism can be used to fabricate gas sensors, as the conductivity of defective CNTs affected by the CA. Furthermore, the obtained consistency of DFTB and DFT calculations has proved the reliability of the highly efficient DFTB method. This would be helpful for us to conduct large-scale calculations and extend research to more complicated systems.

Data accessibility. Additional data are available in the electronic supplementary material.

Authors' contributions. D.L. and F.W. performed simulations. Z.W. and R-Q.Z. conceived this project. D.L., Z.Z., W.J. and Y.Z. analysed results. D.L. produced the figures. D.L., Z.Z., R-Q.Z. and Z.W. wrote the paper. All authors commented on the manuscript.

Competing interests. We declare we have no competing interests.

Funding. The work described in this paper is supported by grants from the National Science Foundation of China (under grant no. 11674123).

Acknowledgements. We thank Miss Sonam Wangmo, Mr Jianpeng Wang, Weiyu Xie, Drs Yan Meng, Minsi Xin, Ruixia Song and Bolong Huang for the stimulating discussions. We also acknowledge the High Performance Computing Center (HPCC) of Jilin University for computation resources.

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
