## [Reviewer comments · Royal Society Open Science]

Review History

RSOS-190727.R0 (Original submission)

Review form: Reviewer 1

Is the manuscript scientifically sound in its present form?

Yes

Are the interpretations and conclusions justified by the results?

Yes

Is the language acceptable?

Yes

Is it clear how to access all supporting data?

Yes

Do you have any ethical concerns with this paper?

No

Have you any concerns about statistical analyses in this paper?

Yes

Recommendation?

Accept with minor revision (please list in comments)

Comments to the Author(s)

This article deals with molecular adsorption into various defects of (5,5) carbon nanotubes by a tight-binding density functional theory method (DFTB-D). The method and the derived results are timely important to the community. The significance is the ration of the unexpectedly high energy of adsorption of the NH₃ molecule found in experimental studies. The results are also interesting because they are a good testing of applicability of DFTB-D approach in the considered tasks. The paper meets the journal requirements and merits publication. However, this referee has queries for consideration:

I am curious if the authors have found difference and any comments on:
discrimination of the electron lone pairs (H₂O and NH₃) from protons (CH₄) interacting with the C dangling bond electrons on the local density of states.

2. Proton interaction with the unpaired electrons of the defect sites

Review form: Reviewer 2

Is the manuscript scientifically sound in its present form?

Yes

Are the interpretations and conclusions justified by the results?

Yes

Is the language acceptable?

Yes

Is it clear how to access all supporting data?

Yes

Do you have any ethical concerns with this paper?

No

Have you any concerns about statistical analyses in this paper?

No

Recommendation?

Accept with minor revision (please list in comments)

Comments to the Author(s)

The manuscript discusses the calculation of binding energies of some small molecules to SWCNT surfaces and surface defects within the DFTB method including empirical London dispersion

forces. The manuscript is well written, interesting and can be published with minor modifications.

Specifically, the authors mention that research efforts into the physical adsorption of small molecules such as CH₄, H₂O and NH₃ on defective nanotubes are seldomly reported. However, the authors might want to comment on a comparison of their study with experimental work by Ulbricht et al. (Carbon 44 (2006) 2931-2942) which reports on experimental determinations of binding energies of all of these molecules and several more on pristine sp² hybridized graphite (as reference material) as well as on the thermal desorption of these molecules from nanotube samples which presumably also includes defects.

In addition, the authors should comment on how their calculations compare with existing experimental adsorption data for small molecule adsorption on graphite which may be used as benchmark for testing the utility of the London corrections.

Decision letter (RSOS-190727.R0)

01-Jul-2019

Dear Dr Wang:

Title: The nature of small molecules adsorbed on defective carbon nanotubes
Manuscript ID: RSOS-190727

Thank you for submitting the above manuscript to Royal Society Open Science. On behalf of the Editors and the Royal Society of Chemistry, I am pleased to inform you that your manuscript will be accepted for publication in Royal Society Open Science subject to minor revision in accordance with the referee suggestions. Please find the reviewers' comments at the end of this email.

The reviewers and handling editors have recommended publication, but also suggest some minor revisions to your manuscript. Therefore, I invite you to respond to the comments and revise your manuscript. I apologise this has taken longer than usual.

Because the schedule for publication is very tight, it is a condition of publication that you submit the revised version of your manuscript before 10-Jul-2019. Please note that the revision deadline will expire at 00.00am on this date. If you do not think you will be able to meet this date please let me know immediately.

Best wishes,
Dr Laura Smith
Publishing Editor, Journals

On behalf of the Subject Editor Professor Anthony Stace and the Associate Editor Professor Tobias Hertel.

RSC Associate Editor:
Comments to the Author:
(There are no comments.)

RSC Subject Editor:
Comments to the Author:
(There are no comments.)

Reviewer comments to Author:

Reviewer: 1

Comments to the Author(s)

This article deals with molecular adsorption into various defects of (5,5) carbon nanotubes by a tight-binding density functional theory method (DFTB-D). The method and the derived results are timely important to the community. The significance is the ration of the unexpectedly high energy of adsorption of the NH₃ molecule found in experimental studies. The results are also interesting because they are a good testing of applicability of DFTB-D approach in the considered tasks. The paper meets the journal requirements and merits publication. However, this referee has queries for consideration:

I am curious if the authors have found difference and any comments on:

discrimination of the electron lone pairs (H₂O and NH₃) from protons (CH₄) interacting with the C dangling bond electrons on the local density of states.

2. Proton interaction with the unpaired electrons of the defect sites

Reviewer: 2

Comments to the Author(s)

The manuscript discusses the calculation of binding energies of some small molecules to SWCNT surfaces and surface defects within the DFTB method including empirical London dispersion forces. The manuscript is well written, interesting and can be published with minor modifications.

Specifically, the authors mention that research efforts into the physical adsorption of small molecules such as CH₄, H₂O and NH₃ on defective nanotubes are seldomly reported. However, the authors might want to comment on a comparison of their study with experimental work by Ulbricht et al. (Carbon 44 (2006) 2931-2942) which reports on experimental determinations of binding energies of all of these molecules and several more on pristine sp² hybridized graphite (as reference material) as well as on the thermal desorption of these molecules from nanotube samples which presumably also includes defects.

In addition, the authors should comment on how their calculations compare with existing experimental adsorption data for small molecule adsorption on graphite which may be used as benchmark for testing the utility of the London corrections.

Author's Response to Decision Letter for (RSOS-190727.R0)

See Appendix A.

Decision letter (RSOS-190727.R1)

22-Jul-2019

Dear Dr Wang:

Title: The nature of small molecules adsorbed on defective carbon nanotubes
Manuscript ID: RSOS-190727.R1

It is a pleasure to accept your manuscript in its current form for publication in Royal Society Open Science. The chemistry content of Royal Society Open Science is published in collaboration with the Royal Society of Chemistry.

On behalf of the Subject Editor Professor Anthony Stace and the Associate Editor Professor Tobias Hertel.

RSC Associate Editor
Comments to the Author:
(There are no comments.)

Reviewer(s)' Comments to Author:

Appendix A

Response to Reviewer' comments on manuscript

RSOS-190727

We are very grateful to the Reviewers for their constructive comments which have been used to improve our manuscript. Listed below are the changes made and responses to the Reviewers' comments.

The major modifications:

1. Line 12 of page 3, added “It should be notice that, for all the adsorbates, it is the proton instead of the heavy atom (N or O) that is pointing to the CNTs. This is because that their electrons are sp^3 hybridized, indicating that electron lone pairs exist in the “proton-free” directions. Such electron-negative lone pair would repulse either with the lone pairs of the dangling C or the π electrons, causing instability of the system, contrary to the electron-relative protons.”
2. Line 60 of page 4: added “To verify the differences of this phenomenon, further electronic density and the local density of state (LDOS) analyses are carried out. It turns out that not only the electron density of the H_2O adsorption in the intermolecular area are obvious larger than those NH_3 and CH_4 , but its LDOS also presents delocalization of the orbitals through the region. This could be concluded that the interaction feature of this system is not simply van de Waals intermolecular interaction. (detailed information is included in SI)”
3. Line 7 of page 5: added “Actually, the activation energy of desorption and the binding energies for small molecules adsorbed on carbon based materials has been systematically observed in the previous experimental researches⁴⁸ The binding energy of CH_4 with SWCNT is of similar magnitude to that of the physical adsorption cases above. This could also prove the reliability of our results.”
4. Added the reference [Ulbricht H., et al. Carbon 2006, 44, 2931-2942.] as Ref.48.

5. In supporting information, we added a new part as:

Part 9. Electronic-based analysis of H₂O, NH₃ and CH₄ adsorbed on the type “a” defected CNT.

Fig. S12 The line electronic density of the H₂O, NH₃ and CH₄ adsorbed on the type “a” CNT. The horizontal axis “relative intermolecular distances” means the geometrical position between the proton of adsorbates and the dangling C atom on the CNT. Due to the fact that the distances between the proton and dangling C atom are different, we normalized the distances into the same scales for comparison.

Fig. S13 The LDOS of the H₂O, NH₃ and CH₄ adsorbed on the type “a” CNT. The horizontal axis is the energy scale of the orbitals, and the vertical axis means the positions from the dangling carbon to the proton. The dangling carbon locates on the origin.

6. In supporting information, we added Part 10 and Part11 providing the stable

structure data for “a” “b” “c” “d” defective and perfect nanotubes adsorbed with small molecules calculated with DFTB-D in the manuscript.

For clarity, we have highlighted all the changes in blue in the main text and supporting information.

REVIEWER REPORT(S):

Reviewer(s)' Comments to Author:

Reviewer: 1

Recommendation: The paper meets the journal requirements and merit publication.

This article deals with molecular adsorption into various defects of (5,5) carbon nanotubes by a tight-binding density functional theory method (DFTB-D). The method and the derived results are timely important to the community. The significance is the ration of the unexpectedly high energy of adsorption of the NH₃ molecule found in experimental studies. The results are also interesting because they are a good testing of applicability of DFTB-D approach in the considered tasks. The paper meets the journal requirements and merit publication. However, this referee has queries for consideration

[Comment] I am curious if the authors have found difference and any comments on discrimination of the electron lone pairs (H₂O and NH₃) from protons (CH₄) interacting with the C dangling bond electrons on the local density of states.

2. Proton interaction with the unpaired electrons of the defect sites.

[Response] Thanks for the constructive comment. However, although the previous geometry optimization with different initial structures has been adopted, only those results with O-H or N-H pointing to the dangling carbon atom on the CNT could be obtained. The lone pairs of the O and N atom would repulse with that of the carbon atom, indicating that such structures would not stably exist.

For exploring the cases of proton interacting with defect sites, we take those with type “a” CNT as example, which contain relative stronger adsorption of H₂O. Both electronic density and the local density of state analyses are carried out. As shown in Fig. R1, the electronic densities on the line of the proton on small molecules and the dangling carbon atom on CNT. The result shows that the electronic density of H₂O is obvious larger than those of NH₃ and CH₄, and meanwhile the density of NH₃ is

slightly larger than that of the CH₄ in the intermolecular area. The extra-large density for H₂O cases corresponds to the result of the charge transfer in Fig. 4 in the main text, indicating a relatively stronger interaction. Similar conclusions could be drawn from the results of local density of states. In Fig. R2, the densities of H₂O system orbitals present a more delocalized feature, especially the orbital around -4 eV, whose density cuts through the intermolecular area. Meanwhile, for the NH₃ and CH₄ cases, the densities of the orbitals are rather localized at each side, corresponding to the conclusion above.

Fig. R1 The line electronic density of the H₂O, NH₃ and CH₄ adsorbed on the type “a” CNT. The horizontal axis “relative intermolecular distances” means the geometrical position between the proton of adsorbates and the dangling C atom on the CNT. Due to the fact that the distances between the proton and dangling C atom are different, we normalized the distances into the same scales for comparison.

Fig. R2 The LDOS of the H₂O (a), NH₃ (b) and CH₄ (c) adsorbed on the type “a” CNT. The horizontal axis is the energy scale of the orbitals, and the vertical axis means the positions from the dangling carbon to the proton. The dangling carbon locates on the origin.

For better illustrate these point, following discussions are added into the main text and the supporting information as: It should be notice that, for all the adsorbates, it is the proton instead of the heavy atom (N or O) that is pointing to the CNTs. This is because that their electrons are sp³ hybridized, indicating that electron lone pairs exist in the “proton-free” directions. Such electron-negative lone pair would repulse either with the lone pairs of the dangling C or the π electrons, causing instability of the system, contrary to the electron-relative protons.” in page 3;

“To verify the differences of this phenomenon, further electronic density and the local density of state (LDOS) analyses are carried out. It turns out that not only the electron density of the H₂O adsorption in the intermolecular area are obvious larger than those NH₃ and CH₄, but its LDOS also presents delocalization of the orbitals through the region. This could be concluded that the interaction feature of this system is not simply van de Waals intermolecular interaction.” in page 4.

Referee: 2

Recommendation: The manuscript is well written, interesting and can be published with minor modifications.

The manuscript discusses the calculation of binding energies of some small molecules to SWCNT surfaces and surface defects within the DFTB method including empirical London dispersion forces. The manuscript is well written, interesting and can be published with minor modifications.

[Comment1]

Specifically, the authors mention that research efforts into the physical adsorption of small molecules such as CH₄, H₂O and NH₃ on defective nanotubes are seldomly reported. However, the authors might want to comment on a comparison of their study with experimental work by Ulbricht et al. (Carbon 44 (2006) 2931-2942) which reports on experimental determinations of binding energies of all of these molecules and several more on pristine sp² hybridized graphite (as reference material) as well as on the thermal desorption of these molecules from nanotube samples which presumably also includes defects.

[Response] Thank you for the comments and suggestions.

We have made the comparison of the experimental work done by the Ulbricht et al. [Carbon 44 (2006) 2931-2942] which reported the interaction of 23 gases and solvents with the basal plane highly oriented pyrolytic graphite (HOPG) with single-wall carbon nanotube samples studied using thermal desorption spectroscopy. This article is necessary for the experimental support of our results, especially for the physical adsorption discussions. The binding energy observed in the experiment is of similar magnitude to those of our calculated results. Relative discussions are added in page 5 as:

“Actually, the activation energy of desorption and the binding energies for small molecules adsorbed on carbon-based materials has been systematically observed in the previous experimental researches. [ref.48] The binding energy of small molecules with SWCNT is of similar magnitude to that of the physical adsorption cases above. This could also prove the reliability of our results.”

[Comment2] In addition, the authors should comment on how their calculations compare with existing experimental adsorption data for small molecule adsorption on graphite which may be used as benchmark for testing the utility of the London corrections.

[Response] Thank you for the comments and suggestions.

As described in the experimental work by Ulbricht et al. [Carbon 44 (2006) 2931-2942]. The interaction of the small molecules with graphitic surfaces is commonly thought to be prototypical of physical interactions. The binding energies of the CH₄, NH₃, and H₂O with the graphitic surface are 17KJ/mol, 25KJ/mol, and 46kJ/mol respectively which can be get in Fig.7 and Fig.8 The experiment data are similar with 100meV around for CH₄ and NH₃, 321meV for H₂O in the manuscript for physical adsorption. The consistency between the experimental data with manuscript can be used as benchmark for testing the utility of the London corrections.

Special thanks for these reviewers' comments and suggestions.